# Two distinct archaeal type IV pili structures formed by proteins with identical sequence

Junfeng Liu[1,4], Gunnar N. Eastep[2,4], Virginija Cvirkaite-Krupovic[1], Shane T. Rich-New [2], Mark A. B. Kreutzberger[3], Edward H. Egelman [3] ✉, Mart Krupovic [1] ✉ & Fengbin Wang [2,3] ✉

Type IV pili (T4P) represent one of the most common varieties of surface appendages in archaea. These filaments, assembled from small pilin proteins, can be many microns long and serve diverse functions, including adhesion, biofilm formation, motility, and intercellular communication. Here, we determine atomic structures of two distinct adhesive T4P from *Saccharolobus islandicus* via cryo-electron microscopy (cryo-EM). Unexpectedly, both pili were assembled from the same pilin polypeptide but under different growth conditions. One filament, denoted mono-pilus, conforms to canonical archaeal T4P structures where all subunits are equivalent, whereas in the other filament, the tri-pilus, the same polypeptide exists in three different conformations. The three conformations in the tri-pilus are very different from the single conformation found in the mono-pilus, and involve different orientations of the outer immunoglobulin-like domains, mediated by a very flexible linker. Remarkably, the outer domains rotate nearly 180° between the mono- and tri-pilus conformations. Both forms of pili require the same ATPase and TadC-like membrane pore for assembly, indicating that the same secretion system can produce structurally very different filaments. Our results show that the structures of archaeal T4P appear to be less constrained and rigid than those of the homologous archaeal flagellar filaments that serve as helical propellers.

Surface appendages such as archaeal type IV pili (T4P) and flagellar filaments are among the most common structures observed in archaea[1–3]. From a structural standpoint, the building blocks of archaeal T4P and flagellar filaments are remarkably similar: both possess a long helical domain at the N-terminus and an immunoglobulin (Ig)–like domain at the C-terminus[1,4–9]. Archaeal T4P are extremely diverse and have been implicated in a variety of functions, including adhesion to biotic and abiotic surfaces, biofilm formation, twitching motility, DNA exchange, and intercellular communication[10,11]. Archaeal flagellar filaments, in all likelihood, have evolved from a subset of archaeal T4P[1], with the major difference between T4P and archaeal flagella being that the latter supercoil, whereas the former do not.

Archaeal flagellar filaments adopt a supercoiled waveform, behaving like Archimedean screws when rotated by a motor within the secretion system[12–14], and thus are essential to the swimming motility of archaea[4]. Key to the supercoiling of archaeal flagellar filaments is the exact helical twist of 108.0°, which aligns ten inner helices vertically after every three turns (1,080°), while archaeal T4P have been observed to have a diversity of helical symmetries, with twist values from -101° to 109°. A comparable observation was made in bacteria, where flagellar filaments all have a helical twist of ~65.5°, thereby aligning eleven inner domains vertically every two turns (720°). In contrast, bacterial T4P, which have no homology with bacterial flagellar filaments, exhibit a large diversity of helical symmetries[1].

[1]Institut Pasteur, Université Paris Cité, Archaeal Virology Unit, Paris, France. [2]Department of Biochemistry and Molecular Genetics, University of Alabama at Birmingham, Birmingham, AL, USA. [3]Department of Biochemistry and Molecular Genetics, University of Virginia School of Medicine, Charlottesville, VA, USA. [4]These authors contributed equally: Junfeng Liu, Gunnar N. Eastep. ✉e-mail: egelman@virginia.edu; mart.krupovic@pasteur.fr; jerrywang@uab.edu

However, outside of the T4P, a large and growing number of examples from both x-ray crystallography and cryo-electron microscopy (cryo-EM) show that the same protein can exist in multiple conformations within the same assembly. Perhaps the most well-known examples involve quasi-equivalence and the deviation from quasi-equivalence in icosahedral viruses. The capsids of SV40 and murine polyoma virus, for example, exist as all pentamers, even at six-coordinated vertices, and this is accommodated by flexible arms that extend from the subunit in different conformations[15]. Similarly, in the bacterial flagellar motor, where there is a symmetry mismatch between the MS-ring and the C-ring, a cryo-EM study has shown that this is accommodated by domain rearrangements in the proteins that form these rings[16]. The two domains present in all T4P, whether bacterial or archaeal, are the conserved N-terminal α-helical domain and a globular non-conserved C-terminal domain. The presence of a short flexible linker between these two domains thus allows for potential domain rearrangements, but these have not previously been observed.

Advances in cryo-EM have dramatically facilitated the determination of atomic structures for biological assemblies, including protein polymers[17]. In this study, we present the high-resolution cryo-EM structures of the *Saccharolobus islandicus* REY15A mono-pilus and tri-pilus and demonstrate that archaeal type IV pilins can also polymerize into dramatically different filament structures. The two pili are assembled from the same pilin polypeptide with seemingly identical glycosylation occupancies and are secreted through the same secretion system. The mono-pilus resembles the previously reported archaeal T4P structures[1,5,6], while the tri-pilus involves a nearly 180° rotation of the Ig-like domain with respect to the mono-pilus conformation. The asymmetric unit in the tri-pilus is three subunits, each with a different orientation of the outer Ig-like domain. The mono- and tri-pili are differentially produced under different growth conditions, suggesting that they have different biological functions.

## Results

### Cryo-EM of mono- and tri-pilus in *S. islandicus* REY15A

For cryo-EM analysis, extracellular appendages were sheared off from the *S. islandicus* REY15A cells by mechanical forces. The cells grown in rich medium (MTSVY), produced a single variety of T4P which were very similar to the adhesive Aap pili previously reported for *S. islandicus* LAL14/1[6], with a slightly improved resolution (3.9 Å versus 4.1 Å). We denote this type of T4P as mono-pilus (see below). The mono-pilus has an asymmetrical unit containing a single pilin with a helical rise of ~4.95 Å and a helical rotation of ~104.8°, displaying a slightly right-handed 7-start feature, similar to the Aap pilus of *S. islandicus* LAL14/1 (Fig. 1c, e, g, Table 1). Notably, *S. islandicus* REY15A has two identical pilin genes, SiRe_2654 and SiRe_2659, located in the vicinity of each other, separated by several transposase pseudogenes and a putative chromosome partition protein Smc (SiRe_2655). We used Sanger sequencing to confirm that SiRe_2654 and SiRe_2659 and even their promoter sequences are indeed identical (Supp Fig. 1). In contrast, *S. islandicus* LAL14/1 has two pilin genes, Sil_2603 and Sil_2606, that are 94% identical in amino acid sequence[6]. The REY15A mono-pilus structure adopts the known archaeal T4P two-domain fold, featuring a long N-terminal α-helix and a C-terminal globular Ig-like β-strand-rich domain[5,6] (Fig. 1c, e). Given the 96% sequence identity shared between the *S. islandicus* REY15A and LAL14/1 pilin protein sequences, the overall structural similarity was unsurprising.

However, when *S. islandicus* REY15A cells were cultured in a poorer medium (MTSV) devoid of yeast extract, in addition to the mono-pilus, we observed the previously published archaeal flagellar filament[4] and a third, structurally dissimilar type of filaments, which we denote as the tri-pilus (Fig. 1a, b, d). The possible symmetries of this structure were calculated based on an analysis of the power spectra. Intriguingly, different helical symmetries were apparent for the inner helical domains and the outer Ig-like domains. The inner long-helix

domain exhibited a rise of 5.27 Å and a twist of 106.8°, similar but not identical to the symmetry for the mono-pilus (4.95 Å, 104.8°). In contrast, the outer domains formed trimers, breaking the symmetry of the inner domain where every subunit was equivalent, and had three times the inner symmetry with a rise of 15.8 Å and a twist of −39.6° (Fig. 1d, f, h). This inner-outer domain symmetry difference has not been previously observed for T4P, but has been reported in some bacterial flagellar filaments (structurally unrelated to T4P), such as enteropathogenic *Escherichia coli* H6, enterohemorrhagic *E. coli* H7, and *Achromobacter*[18], where the globular outer domains can dimerize or tetramerize while the inner coiled-coil domains maintain the symmetry found in other bacterial flagellar filaments where every subunit is equivalent. The final resolution of the tri-pilus reached 3.5 Å, as judged by map:map Fourier shell correlation (FSC), map:model FSC, and $d_{99}$[19]. (Supp Fig. 2, Table 1). At this resolution, direct protein identification from the cryo-EM map can be done unambiguously. We considered the possibility that the different structure of the tri-pilus was due to a different protein than that found in the mono-pilus. We employed two different approaches for protein identification: (1) DeepTracer-ID[20], which uses a combination of AlphaFold[21] predictions and cryo-EM density; and (2) a direct guess of the amino acid sequence from the cryo-EM map using deep-learning methods ModelAngelo[22] or DeepTracer[23], followed by a BLAST search against the *S. islandicus* REY15A proteome. Strikingly, both approaches unequivocally identified the same pilin found in the mono-pilus, seamlessly fitting into the cryo-EM density without any map-model conflicts, such as unexplained side-chain densities (Fig. 2a, b). Other top protein hits from the BLASTP search were thoroughly examined and confirmed to not align with the cryo-EM map (Supp Fig. 3). Both mono- and tri-pilin structural models suggest that the first 12 residues correspond to the signal peptide, which is cleaved from the mature pilin protein (Lys11-Ala12-↓-Leu13-Ser14), consistent with the previously established recognition motif of the signal peptidase PibD[2]. Notably, semiquantitative comparison of the abundance of mono-pili versus tri-pili showed that when the cells were grown in the MTSV medium, tri-pili were 6.5× more abundant than mono-pili.

### Mono-pilin versus tri-pilin

Both mono- and tri-pilin structures consist of a long N-terminal α-helix coupled with a C-terminal globular β-strand-rich domain (Fig. 2a, b). When one of these two structural domains is aligned between mono- and tri-pilins, the domain architectures appear largely consistent with an RMSD between 0.3 and 0.9 Å (Supp Fig. 4). However, compared to the mono-pilin, the globular domain of tri-pilins undergoes a significant conformational change. Rather than projecting out from the N-terminal helix, it pivots almost 180° and bends towards the N-terminal helix (Fig. 2c). The relative orientation of the globular domain with respect to the N-terminal helix also varies across the three different subunits, A, B, and C, in the tri-pilus asymmetric unit. These different conformations are facilitated by a flexible Gly-Gln-Gly linker region situated between the two domains, and the cryo-EM densities for this linker region are clearly seen (Supp Fig. 5). Currently, it remains unclear whether those structurally different mono- and tri-pilins are products of one of the two or both identical genes, SiRe_2654 and SiRe_2659.

### Mono- and tri-pilus display heavy glycosylation at identical sites

We asked whether the distinct mono-pilus and tri-pilus formations could arise from variations in the levels or patterns of post-translational modifications, particularly glycosylation. A previous study reported that glycosylation accounts for more than 35% of the total molecular weight in the *S. islandicus* LAL14/1 pilus[6]. Considering that the protein sequence of *S. islandicus* REY15A pilin is nearly identical to that of *S. islandicus* LAL14/1 pilin, we anticipated a high degree of glycosylation in both the mono-pilus and tri-pilus of *S. islandicus*

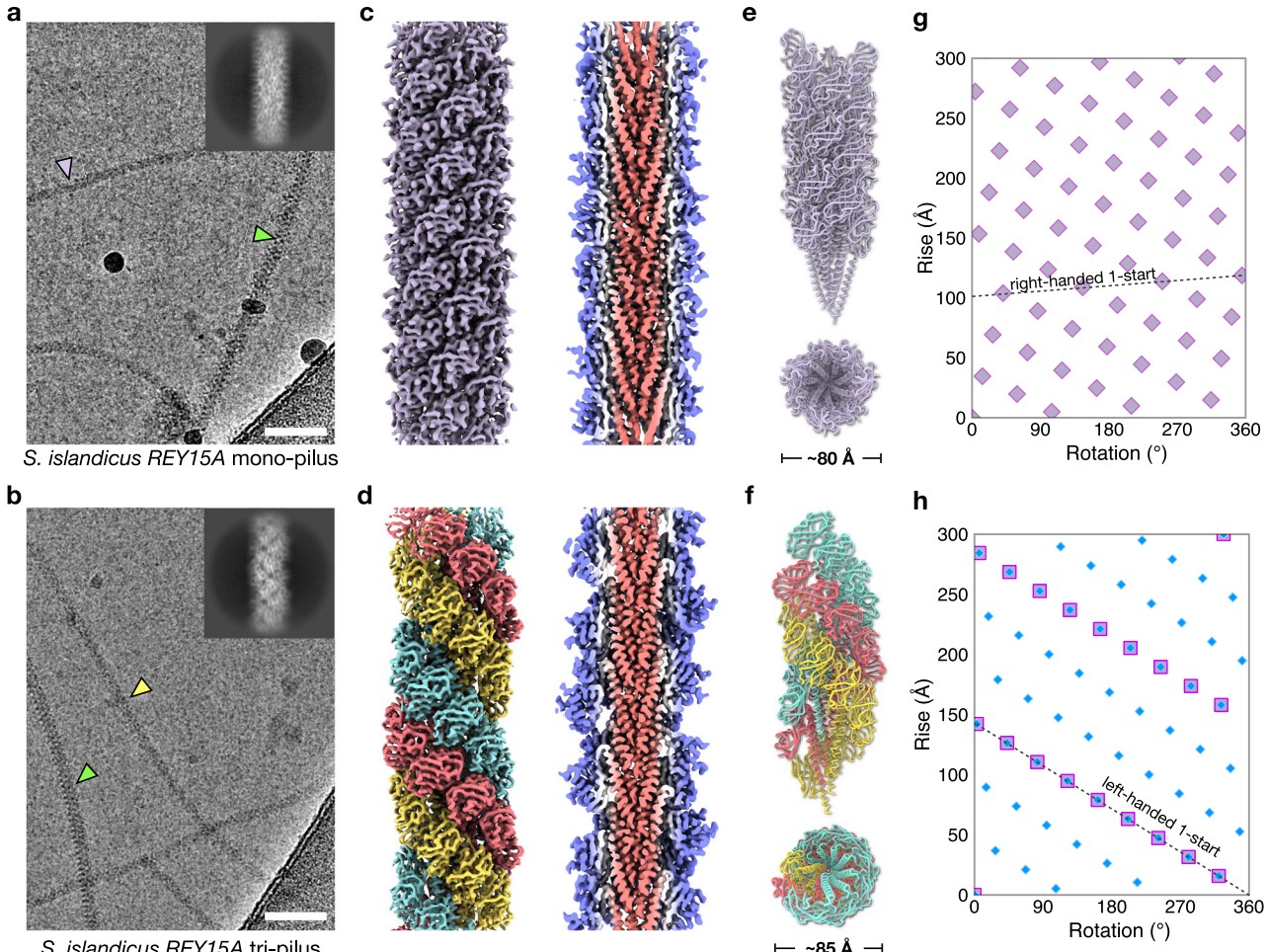

**Fig. 1 | Cryo-EM of the S. islandicus REY15A mono-pilus and the tri-pilus.**
Representative cryo-electron micrographs, taken from 1,032 micrographs recorded, of the *S. islandicus* REY15A mono-pilus **a** and tri-pilus **b**. Scale bar, 50 nm in both **a** and **b**. Lavender arrowhead points to the *S. islandicus* REY15A mono-pilus; green arrowheads point to the *S. islandicus REY15A* flagellar filament; yellow arrowhead points to the *S. islandicus* REY15A tri-pilus. 2D averages of mono-pilus and tri-pilus are shown in the top right corner. Cryo-EM reconstructions of the *S. islandicus* REY15A mono-pilus at 3.9 Å resolution **c** and the tri-pilus at 3.5 Å resolution **d**). For the mono-pilus, all subunits are lavender and all are equivalent; for the tri-pilus, the three subunits in the asymmetric unit are colored cyan, red, and yellow. Thin sections parallel to the helical axis of the pilus are shown, colored by the radius. Side and top view of the mono-pilus **e** and tri-pilus **f** atomic models built into the cryo-EM maps shown in **c**, **d**. Models are colored the same as cryo-EM maps. Helical nets of the mono-pilus **g** and tri-pilus **h** using the convention that the surface is unrolled and viewed from the outside. For the mono-pilus, pilin subunits are displayed as diamonds and the right-handed 1-start helix is shown. For the tri-pilus, the asymmetrical units (pilin trimers) of the filament are displayed as squares. The N-terminal α-helices of the pilin, which do not form trimers, are displayed as blue diamonds. The left-handed 1-start relating pilin trimers is shown.

REY15A. Upon comparing the glycosylation sites and levels in mono-pilin and tri-pilins, we found identical O-glycosylated residues - Thr95, Ser97, Thr102, Ser104, and Ser106 - in both mono-pilin and tri-pilins (Fig. 3a, Supp Fig. 6). There are no extra densities observed associated with the four asparagine side chains present in this pilin, consistent with the study of the *S. islandicus* LAL14/1 pilus[6] where N-glycosylation was also not detected. Moreover, the level of extra densities coming from surface glycosylation between mono-pilus and tri-pilus was similar, despite their glycosylation patterns evidently varying due to changes in symmetry (Fig. 3b-c, Supp Fig. 6). It remains possible that glycan species may differ between mono-pilus and tri-pilus, even if their densities appear similar. Therefore, we cannot completely rule out the possibility that glycosylation may be a factor differentiating mono-pilus and tri-pilus formation.

### Plasticity in the packing of inner helices between mono- and tri-pilus

The slight differences in helical parameters of the inner helices between the mono- and tri-pilus raised interesting questions regarding the mechanisms allowing such alternate packing. In a given inner helix

of pilin $S_0$, unique interactions occur with four out of eight total interacting helices, namely, $S_{+1}$, $S_{+3}$, $S_{+4}$, and $S_{+7}$ (Fig. 4a) (by symmetry, identical interactions will be made with $S_{-1}$, $S_{-3}$, $S_{-4}$, and $S_{-7}$). The small helical symmetry difference per subunit, 0.32 Å in rise and 2° in rotation, is considerably amplified when interactions take place between more distant pilins along the 1-start helix. For instance, the interaction between $S_0$ and $S_{+7}$ is predicted to shift from its original position by approximately 2.3 Å in rise and 14° in rotation (Fig. 4a, b) in the inner core between a mono- and tri-pilus, an axial shift equivalent to almost half of the α-helical pitch. As expected, the buried surface area between $S_0$ and $S_{+1}$ remains nearly constant between the mono- and tri-pilus, while the other three interfaces are significantly affected.

On the other hand, the total buried surface due to inner-helix interactions in the core remained largely unaltered between the mono- and tri-pilus. Further analysis of these interfaces revealed that residues participating in these inter-helical interactions are predominantly hydrophobic: only a single threonine and two serine residues were found at the lower part of the helix; no charged residues are involved. The hydrophobic effect, by definition, is nonspecific[24,25]. Hence, the hydrophobic surfaces of these helices can accommodate the different

**Table 1 | Cryo-EM and refinement statistics of mono-pilus and tri-pilus**

| Parameter | Mono-pilus | Tri-pilus |
|---|---|---|
| **Data collection and processing** | | |
| Voltage (kV) | 300 | 300 |
| Electron exposure (e$^-$ Å$^{-2}$) | 50 | 50 |
| Pixel size (Å) | 1.08 | 1.08 |
| Particle images (n) | | |
| Shift (pixel) | 6 | 15 |
| **Helical symmetry** | | |
| Point group | C1 | C1 |
| Helical rise (Å) | 4.95 | 15.80 |
| Helical twist (°) | 104.80 | −39.65 |
| **Map resolution (Å)** | | |
| Map:map FSC (0.143) | 3.9 | 3.5 |
| Model:map FSC (0.5) | 4.0 | 3.6 |
| $d_{99}$ | 4.1 | 3.8 |
| **Refinement and Model validation** | | |
| Ramachandran Favored (%) | 98.5 | 93.8 |
| Ramachandran Outliers (%) | 0.0 | 0 |
| Real Space CC | 0.86 | 0.86 |
| Clashscore | 12.2 | 9.3 |
| Bonds RMSD, length (Å) | 0.006 | 0.005 |
| Bonds RMSD, angles (°) | 0.969 | 0.967 |
| **Deposition ID** | | |
| PDB (model) | 8TIF | 8TIB |
| EMDB (map) | EMD-41286 | EMD-41283 |

packing, facilitating helices to slide past each other while preserving these hydrophobic contacts. We previously applied this concept to the spindle-shaped virus SMV1[26], where hydrophobic surfaces between largely α-helical subunits allow strands of such subunits to slide past each other. Here, we observed another example: the inner helices of the mono- and tri-pilus.

### Inner helices are no longer perfectly buried in the tri-pilus

In contrast to the mono-pilus, where inner helices are fully shielded by outer domains, portions of the inner helices in the tri-pilus become exposed to the solvent (Supp Fig. 7). This observation prompted us to investigate how the interactions between the inner helix and outer domains have changed from the mono- to the tri-pilus. In the mono-pilus, the inner helix of a mono-pilin $S_0$ interacts with outer domains from pilins $S_{+4}$, $S_{+7}$, and $S_{+11}$, resulting in a total buried surface of approximately 873 Å$^2$ (Fig. 4c).

By contrast, in the tri-pilus, such interactions differ among tri-pilins A, B, and C. The inner helix of tri-pilin A interacts with the outer domains of itself $S_0$, $S_{+3}$, and $S_{-7}$, yielding a total buried surface of 350 Å$^2$. For tri-pilin B, the inner helix interacts with the outer domains of itself $S_0$, $S_{+3}$, and $S_{-4}$, with a total buried surface of 500 Å$^2$. The inner helix of tri-pilin C engages with the outer domains of itself $S_0$, $S_{-3}$, and $S_{-4}$, leading to a total buried surface of 403 Å$^2$ (Fig. 4c). Collectively, the data show that all inner helices in the tri-pilus are less shielded by the outer domains compared to those in the mono-pilus, which could have consequences for the pilus stability.

### Trimerization of outer domains in tri-pilus

In the mono-pilus, the outer domains are densely packed. The outer domain of $S_0$ interacts with six other outer domains, exhibiting an extensive total buried surface area of 1,574 Å$^2$. Three out of those six interfaces are unique (with the other three related by symmetry), with

buried surface area ranging from 236 to 300 Å$^2$ (Fig. 5a). Conversely, in the tri-pilus, the outer domains bend away from the helical axis (Fig. 2). When aligning the outer domains between the mono- and tri-pilus, it becomes apparent that similar regions of the outer domain are involved in the contacts. However, the outer domains in the tri-pilus are clearly rotated and translated from their position in the mono-pilus (Fig. 5b). Notably, all outer domains of the tri-pilus exhibit significantly less buried surface area than those of the mono-pilin: 907 Å$^2$ for pilin A; 959 Å$^2$ for pilin B; 842 Å$^2$ for pilin C. In addition, a groove forms between the outer domains of pilin A and pilin C, with a minuscule buried surface area of 41 Å$^2$ between the two outer domains on each side of this groove.

### ATPase SiRe_0181 and TadC-like membrane pore SiRe_0180 are required for both mono- and tri-pilus production

The large difference in orientation and packing between the outer domains of mono- and tri-pilus prompted us to question whether the same secretion system assembles both types of pili or the two types of filaments were produced by different systems. Indeed, the loci encoding the mono/tri-pilin protein and the cognate secretion system are located 192 kbp apart, suggesting that pilins can be secreted in trans (Fig. 6a). *S. islandicus* REY15A encodes four T4P systems, namely, adhesive Aap pilus reported herein, flagellum, UV-inducible pilus and bindosome complex[27]. Each of these systems is associated with a cognate secretion ATPase and TadC-like membrane pore protein (Fig. 6a).

We surmised that if the tri-pilus was assembled in trans by a non-cognate secretion system, knockout of the corresponding ATPase would eliminate the production of tri-pili, while still maintaining the production of mono-pili. Thus, we constructed knockout mutants of all ATPases associated with the T4P systems and analyzed the filaments produced by the corresponding mutants using cryo-EM. The results showed that knockout of ATPase SiRe_0181 eliminated production of both mono- and tri-pili, whereas deletion of the other three ATPase genes had no noticeable impact on pili production (Fig. 6b, Supp Fig. 8). Notably, flagellar filaments were no longer observed not only in the knockout of the flagellar ATPase SiRe_0119, but also upon deletion of the ATPase SiRe_2519 associated with the bindosome complex, suggesting an interplay between the two systems.

This led us to question whether it is the TadC-like membrane pore component, rather than the ATPase, that determines the pilus structure. Thus, we individually deleted the four TadC-like genes (Fig. 6a) and analyzed the production of the filaments by cryo-EM. However, the results echoed the previous observations: the deletion of gene *SiRe_0180*, which is located in the same locus as the ATPase *SiRe_0181*, halts the production of both mono- and tri-pili, whereas the other three knockout mutants still allow the production of tri-pili. Therefore, we conclude that both mono- and tri-pili are assembled by the same secretion system, despite the large domain rearrangements in the outer surface of the two types of filaments.

Finally, we considered the possibility that the difference in pilus structure might be determined by the expression level of the cognate secretion system components. To this end we performed RT-qPCR analysis, which showed that with the exception of the ATPase (SiRe_1879) associated with the UV-inducible pilus, all other ATPase and TadC-like genes were transcribed when the cells were grown in either MTSVY or MTSV medium, albeit at different levels (Fig. 6c). For instance, the level of expression of the flagellar ATPase was considerably lower compared to the other genes, consistent with low level of flagellation in *S. islandicus* REY15A[4]. Importantly, the expression levels of both ATPase and TadC proteins responsible for the production of mono- and tri-pili were significantly different under growth in the MTSV and MTSVY medium, with the expression in rich medium (MTSVY) being nearly twice higher for both genes. For all other ATPase and TadC genes, with potential exception of the ATPase associated

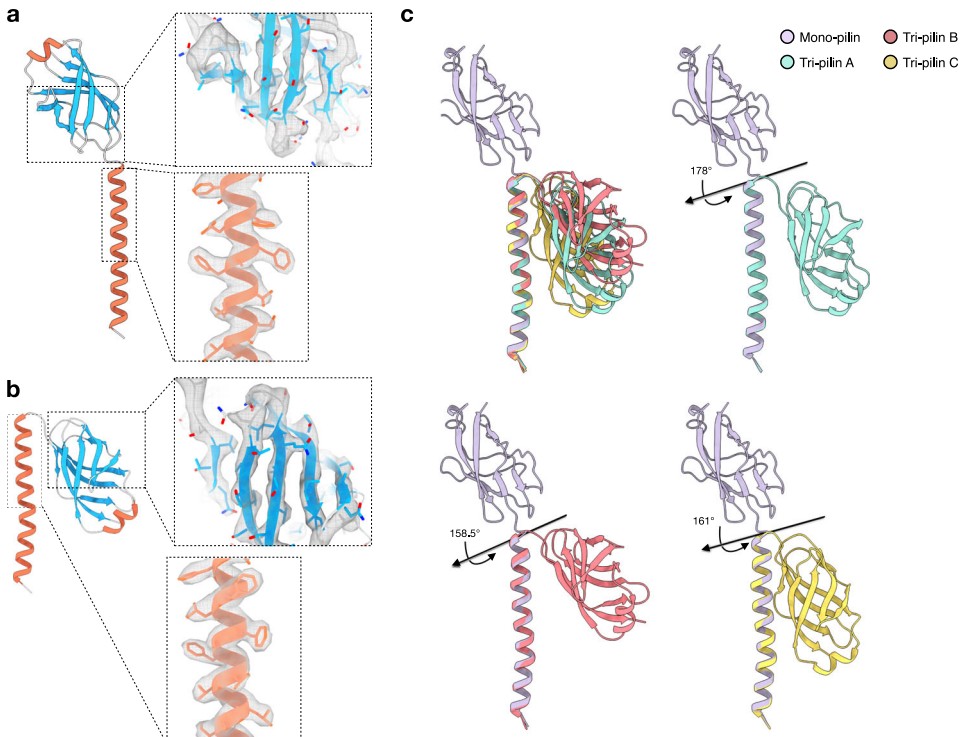

**Fig. 2 | *S. Islandicus REY15A* mono-pilin versus tri-pilins.** Atomic model of mono-pilin **a** and tri-pilin A **b** in cartoon representation with β-sheets colored blue and α-helices colored red. The map quality of the N-terminal long helix and C-terminal Ig-like domain are shown on the right. **c** Alignment of the three pilins of the tri-pilus asymmetric unit (A, cyan; B, red; C, yellow) and the single pilin of the mono-pilus asymmetric unit (purple) by their N-terminal helices is shown. The angles and axes of rotation between each of the tri-pilin C-terminal Ig-like domains and the mono-pilin Ig-like domain are labeled.

with the bindosome complex, there was no significant difference in gene expression under the two growth conditions.

## Discussion

The field of life sciences has been revolutionized by recent advances in cryo-EM[17,28]. Besides offering near-atomic resolution structures of macromolecular assemblies, this technique has proven immensely valuable for the de novo identification and characterization of protein components that make up previously unknown assemblies, with atomic models being directly built into the map densities[29–31]. In this study, we reported the presence of an archaeal type IV pili in two distinct forms: the canonical mono-pilus form, which aligns with previously reported archaeal T4P structures, and the novel tri-pilus form characterized by the outer domain trimerization, a feature not previously observed in other T4Ps. Strikingly, through cryo-EM, we discovered that these two structurally different pili are composed of the same pilin polypeptide. The presence of two filament types derived from a single pilin protein shows that the same protein is able to pack into four different environments: one in the mono-pilus, and three in the tri-pilus. Surprisingly, the outer domains of the mono- and tri-pilus exhibit almost opposite orientations compared to their inner helical domains. Simultaneously with us[32], Gaines et al. report a tri-pilus structure from *Sulfolobus acidocaldarius*, suggesting that this unusual T4P conformation occurs more widely among members of the order Sulfolobales[33,34]. The unexpected diversity of archaeal T4P structures is complemented by the very recent finding that bacterial T4P can also be more diverse than originally assumed[35]. The ability of T4P subunits in both bacteria and archaea to generate diverse structures with diverse properties may help explain why the corresponding genes are so ubiquitous in prokaryotes.

The completely different outer domain conformations strongly suggest that a direct transformation from one pilus form to another is highly unlikely. First, given the small change in twist (~2°) for the inner helices between the two forms, the distal end of a 2 μm-long filament (containing ~4000 subunits) would need to rotate by ~8,000° (~22 turns) in a switch between the two forms. Second, the energy needed to break all of the extensive interfaces between the outer domains in one state to allow new ones to form in the other state would be huge. Third, and most importantly, it would be impossible to flip a single outer domain in the assembled filament from a down orientation (in the tri-pilus) to an up orientation (in the mono-pilus), or vice versa, due to the steric clashes with neighboring outer domains. Somehow, all of the thousands of subunits would need to flip at the same time. A fourth issue is that the tri-pilus conformation extends coherently over long distances. If this trimerization were to occur after filaments were polymerized, we might expect to see random patches of trimers. But we always see the pattern of three different strands in Fig. 1d, and never random patches. It is therefore impossible to imagine how a transformation between the two forms, mono- and tri-pilus, could be done without fully disassembling the pilus.

The key to the large domain rearrangements is the extremely flexible linker between the two domains. It has previously been proposed that this linker is more flexible in archaeal flagellar filaments than it is in archaeal T4P[7]. The present results directly conflict with that suggestion. Further, the linker in all archaeal flagellar filaments is three residues long, while the linkers in archaeal T4P are ~3–10 residues long[1], suggesting that T4P with these longer linkers may also display great flexibility between the two domains. It was previously noted that the helical twist in archaeal T4P ranged from ~101° to 109°[1]. The simplest interpretation of this was that these differences in helical symmetry were due to sequence differences among the T4P. But here we show that the identical sequence can assemble into an inner core with a helical twist of either 104.8° or 106.8°, and thus the precise helical twist is determined by different interactions between the N-terminal domain and the globular domain, as well as by the interactions between the globular domains. Thus, the symmetry relating the α-

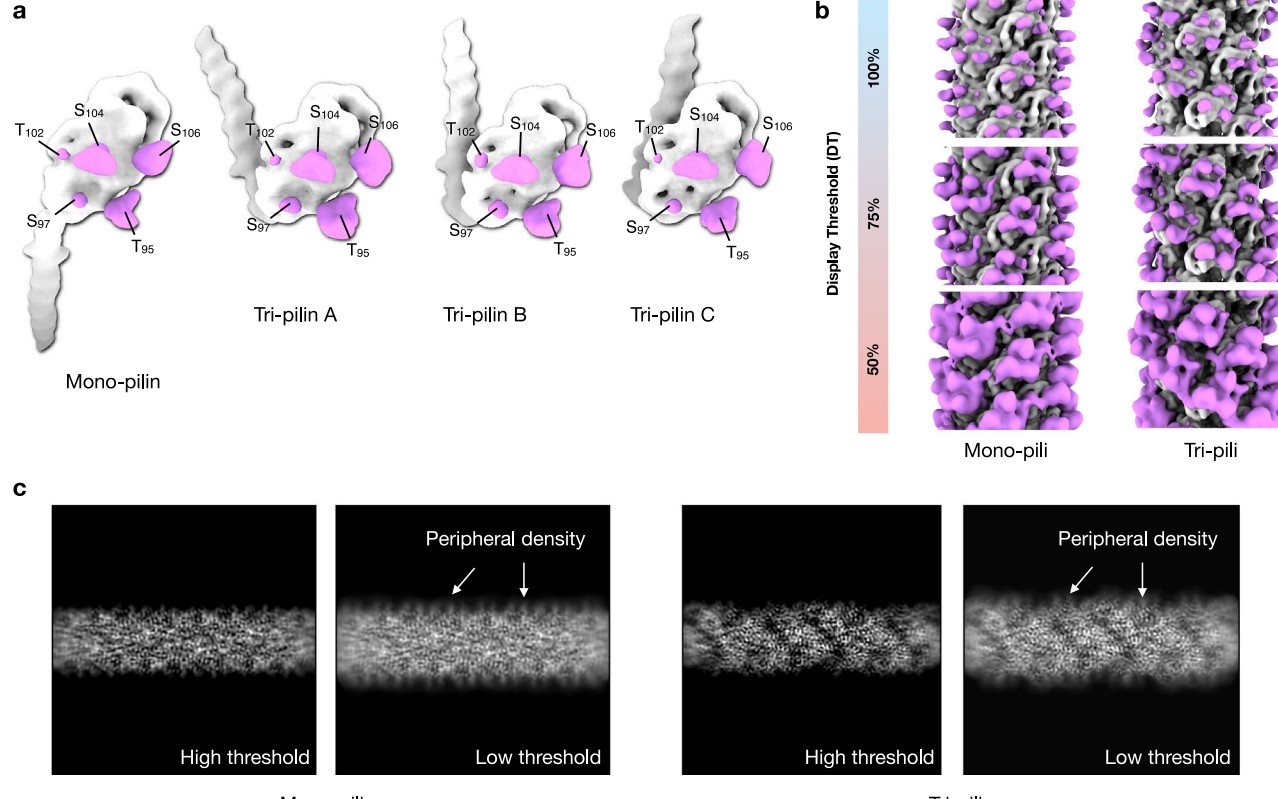

**Fig. 3 | Mono-pilus and tri-pilus show similar levels of surface glycosylation.**
**a** Cryo-EM map densities due to post-translational modifications on pilin subunits. For a clear comparison, all volumes were filtered to 7 Å, and one mono-pilin and three tri-pilins were aligned by the outer domain. The densities accounted for the protein part (grey) were segmented out using a mask generated from the refined atomic model. The extra experimental density after subtracting the protein part, presumably comes from O-linked sugars on Ser or Thr residues, are colored in magenta. The amino acid side chains with extra densities are labeled. **b** Surface glycosylation of the mono-pilus (left) and tri-pilus (right). The density accounted for by protein models is colored in gray, and the extra density is colored in magenta. Three different display thresholds for the putative surface glycosylation are shown. The 100% display threshold is arbitrary chosen by the best visual display of the map of the protein part. All volumes were lowpass filtered to 7 Å for a clear comparison. **c** Projections of the 3D reconstruction of mono-pilus (left) and tri-pilus (right) at low and high threshold. Fuzzy coats were seen at low threshold clearly.

helical core domains of archaeal T4Ps might be considered malleable, with the plasticity imparted by the hydrophobic helix-helix interactions. This is not unlike the interactions observed in the lemon-shaped archaeal virus SMV1[26]. However, the local packing interactions in different symmetries of SMV1 are nearly identical, while the core helix interactions between the tri- and mono-pilus are considerably different along the 7-start helix.

The atomic models we have constructed for mono- and tri-pilus revealed similar but not identical inner helix packing, yet distinctly different outer domain arrangements. Given that the conformational change between preassembled mono- and tri-pilus is highly unlikely, we considered whether the two types of pili were assembled via the same or distinct secretion systems. Our results unequivocally show that the same secretion system is responsible for the production of both filaments, with knockouts of either the cognate ATPase or TadC-like gene completely eliminating the production of both mono- and tri-pili, indicating that these two pili are likely produced by the same secretion system. From a structural standpoint, both mono- and tri-pilus share a similar inner helix core, suggesting that structural differences within a few pilin subunits may be tolerated by the platform proteins aligning them before secretion. Despite the slightly larger outer diameter of the tri-pilus due to the Ig-like outer domain rearrangements, both pili forms are coated with a similar level of glycosylation on the surface. Since the globular domain bends towards the N-terminal helix in the tri-pilus, we do know that this conformation must be established after the subunit is extracted from the membrane,

as otherwise the outer domain would be in a steric clash with the membrane.

The fact that mono- and tri-pili are assembled by the same secretion system and share similar inner core helical symmetries is quite striking. We speculate that a key difference between the two forms could be a reduced stability of the tri-pilus. Previous work has demonstrated that a closely related mono-pilus exhibits remarkable stability under extremely harsh conditions, such as boiling in GuCl or pepsin digestion[6]. This can be attributed to the fact that the inner hydrophobic helices of the mono-pilus are completely encased by the outer domains, which are further armored by heavy glycosylation. In contrast, in tri-pilus, the hydrophobic inner helices are not perfectly enveloped by the outer domains, suggesting reduced stability compared to the mono-pilus. Thus, the tri-pilus might offer additional benefits to the cells like enhanced motility due to potentially faster kinetics for disassembly.

Under conditions of nutrient limitation, *S. islandicus* REY15A cells predominantly express tri-pili, although mono-pili and flagella are also present at low levels; however, in the rich medium, the cells exclusively express the mono-pilus. This transition may represent a switch in lifestyle from a sessile, adherent state in rich medium, to a potentially locomotive state in search of a more favorable environment under nutrient limitation conditions. The functions of flagella and adhesive pili are inherently antagonistic and expression of both could lead to futile energy expenditure. Indeed, the production of archaeal flagella is highly costly from a bioenergetic standpoint;[36] hence, flagellar

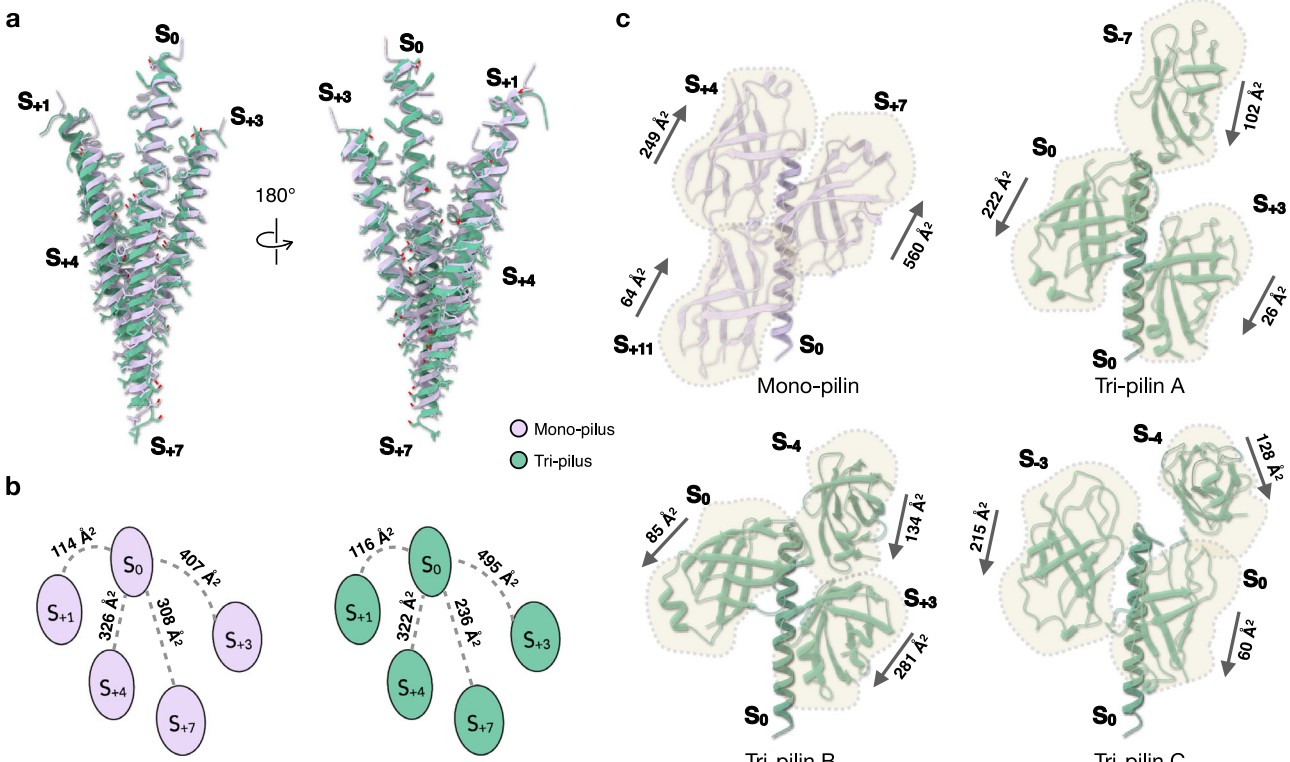

**Fig. 4 | Inner helices of the tri-pilus display altered inner-core packing and become more solvent-accessible than in the mono-pilus. a** For a given pilin $S_0$ inner helix (residue 13–48), eight surrounding helices interact with it. Four unique interactions with $S_{+1}$, $S_{+3}$, $S_{+4}$, and $S_{+7}$ are displayed. Mono-pilins and tri-pilins were colored in lavender and green, correspondingly. Side chains are also shown. **b** Buried surface area (Å²) of unique interactions shown in **a**. **c** Contacts between the inner helix and outer domains. The inner helices of pilin $S_0$ are aligned between mono-pilin and three different tri-pilins. Outer domains from other pilin subunits contacting this inner helix are shown, with the buried surface area labeled. The arrows indicate the direction from the start of the outer domain (Thr49) to the C-terminus of the outer domain (Ser143).

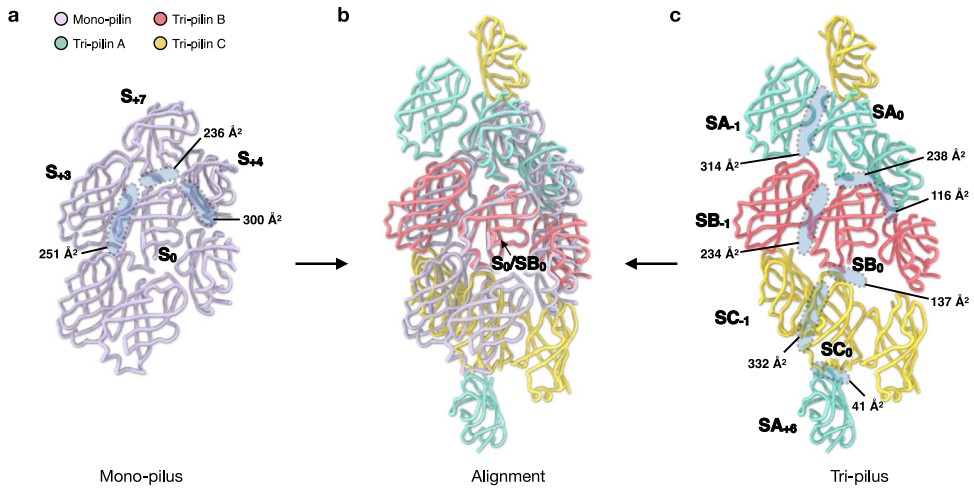

**Fig. 5 | Trimerization of outer domains in the tri-pilus leads to a new type of inter-subunit contact. a** The outer domain interactions of the mono-pilus. $S_0$ is shown in the middle and outer domains of other pilin subunits that interact with it are labeled. The buried surface areas of unique interfaces are labeled. **b** alignment of outer domain lattice of mono- and tri-pilus. **c** The outer domain interactions of the tri-pilus. $SB_0$ (subunit of tri-pilin B) is shown in the middle, and other subunits are included to show all unique contacts. The buried surface areas of unique interfaces are labeled. Three different pilins in tri-pilus are colored consistent with Fig. 1.

filaments are produced only when necessary, whereas under the conditions of adhesive mono-pili expression, no flagella were observed (Fig. 6b). Thus, the adhesive properties of tri-pili might be radically different compared to those of mono-pili, in order to avoid obstructing the flagella-driven swimming motility. Although the molecular basis for the ability of the same secretion system to assemble two very different pili remains unclear, we hypothesize that significantly different expression levels of the secretion system under distinct growth conditions hold the clue to this conundrum. In particular, under the conditions when the adhesive mono-pili are produced, the expression of the cognate secretion ATPase and TadC-like gene is nearly twice higher compared to their expression under the conditions when tri-pili

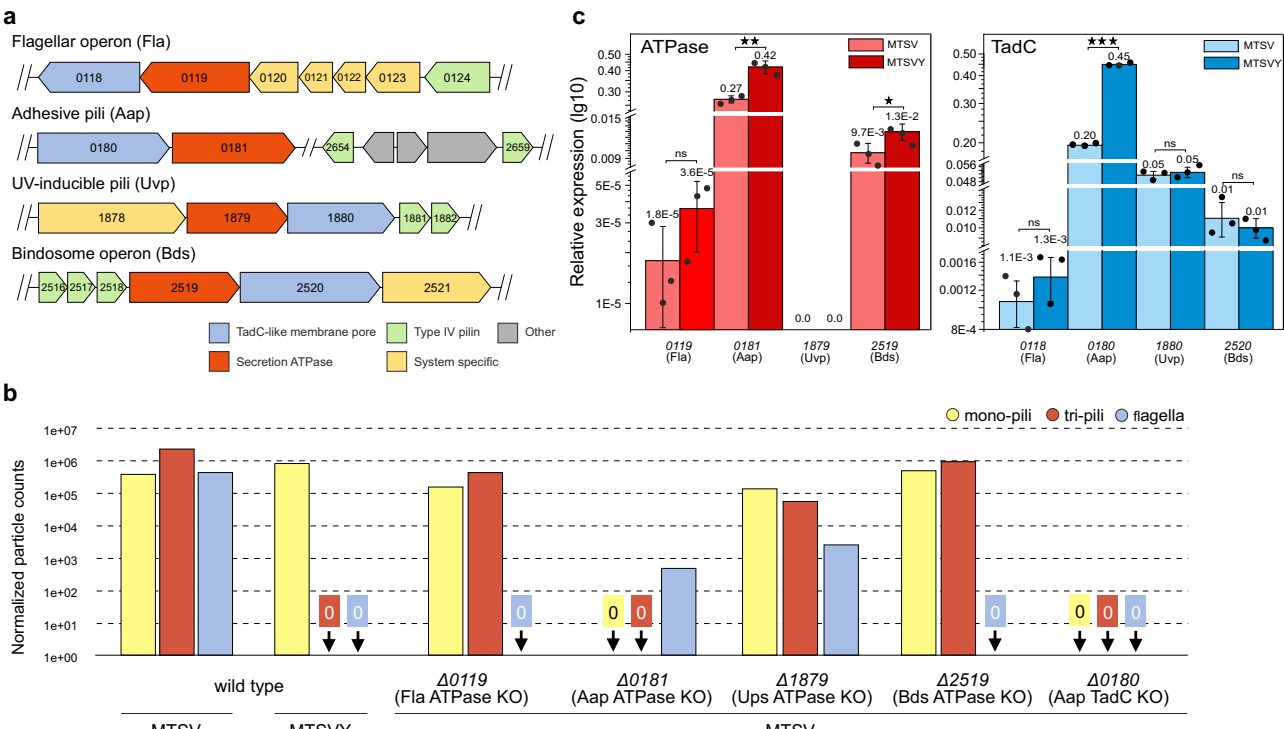

**Fig. 6 | ATPase SiRe_0181 and TadC-like membrane pore SiRe_0180 are required for production of both pili. a** Gene clusters from the *S. islandicus* REY15A genome which contain the pilin proteins and gene clusters containing ATPase and TadC genes are shown. SiRe_2654 and SiRe_2659 (green) have identical translated sequences that correspond to the protein identified in REY15A mono- and tri-pilus filaments. The length of the arrow corresponds to the gene length. **b** The relative filament abundance of three different filaments observed under different medium and ATPase knockout strains. As there is no simple way to identify between mono-pilus and tri-pilus from a single filament on a cryo-EM micrograph, this analysis was done in a semi-quantitative way. A similar amount of cells were collected, and the same shearing and plunge freezing process were performed. After cryo-EM imaging, all filaments were auto-picked with a shift of 5 Å, and 2D classifications were run to count the number of particles from different filament species. After that, the number of particles was normalized against WT-poor conditions in four different factors: camera pixel size; camera width K3 vs. Falcon 4; number of micrographs collected; sample dilution used. The total number of micrographs (N) collected and total counted particles without normalization (P) for each dataset are: (1) WT-poor,

$N = 1,000$, $P = 1,112,051$; (2) WT-rich, $N = 2,700$, $P = 1,731,245$; (3) Δ0119-poor, $N = 1,300$, P = 765,665; (4) Δ0181-poor, $N = 4,500$, $P = 2,396$; (5) Δ1879-poor, $N = 600$, $P = 115,205$; (6) Δ2519-poor, $N = 2,300$, $P = 543,802$; (7) Δ0180-poor, N = 558, $P = 0$. **c** RT-qPCR analysis of the ATPase (left panel) and TadC (right panel) gene expression levels in REY15A cells cultivated in poor-MSTV and rich-MSTVY medium. Cell cultures were sampled after cultivated for 18 h. 16 S rRNA was used as the reference and *tbp*, a housekeeping gene encoding TATA-binding proteins, was used as the control. The transcription levels of tbp were defined as 1.0, and the relative expression levels of the ATPase and TadC genes were calculated based on the transcription level of *tbp*. Three biological replicates were analyzed. The individual data points were overlayed with the bar charts. Error bars represent Standard Deviation from three independent experiments. The mean values were indicated on top of the bars. Two Sample t test was used to evaluate statistical significance. The lines and asterisks on top of the figure denote the pairwise statistical significance, *$p < 0.05$, **$p < 0.01$, ***$p < 0.001$. Source data are provided as a Source Data file.

are dominant (Fig. 6c). It is possible that the lower expression of the secretion system components leads to its inundation with pilins, eventually resulting in different pilus structure, compared to the conditions where the secretion system is more actively expressed. This hypothesis should be tested in the future by fine-tuning the expression of different secretion system components.

The dimerization and tetramerization of the outer domain in cell appendages has also been observed in several bacterial flagellar filaments[18,37–40]. For instance, the outer domains of flagellar filaments from enterohemorrhagic *E. coli* H7 and enteropathogenic *E. coli* H6 can dimerize and tetramerize, respectively, presumably enhancing their swimming capabilities in high-viscosity environments[18]. In *Pseudomonas aeruginosa*, a different form of dimerization of the outer domains was shown to be essential for motility[40]. In the tri-pilus, the outer domains trimerize to form a closely packed left-handed trimer protofilament, with minimal interactions between different turns of the protofilament. Compared to the mono-pilus, the tri-pilus could be stretched more easily by allowing temporary breakage of the interactions between the protofilament turns. The trimerization of the outer domains also results in a significant groove, with mostly uncharged residues present within the groove. While DNA uptake via much larger

positively charged grooves has been shown in some bacterial T4P[41–43], such nucleotide uptake seems unlikely for the tri-pilus due to its surface charge (Supp Fig. 7). However, the groove may facilitate the uptake of other nutrients. We anticipate that additional functional studies of tri-pili will yield more detailed insights.

## Methods

### *S. islandicus* REY15A filament preparation

*S. islandicus* REY15A[44] cells were grown aerobically with shaking (145 rpm) at 75 °C as described previously[4]. The cells were grown either in MSTV medium containing mineral salts (M), 0.2% (wt/vol) sucrose (S), 0.2% (wt/vol) tryptone (T), a mixed vitamin solution (V) or MTSVY medium, which in addition contained yeast (Y) extract (0.1% wt/vol)[45]. We refer the MSTV as poor medium and MSTVY as rich medium in the main text. For isolation of filaments, REY15A cells were inoculate into 300 ml of MTSV or MTSVY medium at an initial $OD_{600}$ of 0.05 and cultivated for 18 h. Once cooled down to room temperature, the cell cultures were vortexed to shear off the filaments from the cell surface. The cells were removed by centrifugation at 7,420 x *g* for 20 min and the filaments were pelleted from the supernatant by ultracentrifugation (Type 45Ti rotor, 185,511 x *g*) at 10 °C for 2 h.

**Generation of gene knockout *S. islandicus* REY15A strains**

The genes encoding the ATPase and TadC proteins indicated in Fig. 6a were knocked out using the endogenous CRISPR-based genome editing system in S. islandicus REY15A[46]. The 40-nt protospacers were selected on the target gene downstream of the 5′-CCN-3′ sequence. Spacer fragments were generated by annealing the corresponding complementary oligonucleotides and inserted into the genome-editing plasmid pGE at the BspMI restriction site. The donor DNA was obtained by overlap extension PCR between the L-arm and R-arm corresponding to the genes of interest and inserted into pGE at the SphI and XhoI restriction sites. Plasmid pGE was then introduced into E233S cells by electroporation and transformants were selected on MSCV plates without uracil (Supp Fig. 9). The gene knockout strains were determined by PCR amplification with flanking and gene-specific primer pairs. The plasmids and strains constructed and used in this study are listed in Supp Tables 1-2. The selected spacers and the oligonucleotide sequences for PCR used in this study are listed in Supp Tables 3-4.

**Quantitative reverse transcription PCR (RT-qPCR)**

RT-qPCR was carried out as previously described[47]. Specifically, REY15A cells were collected after cultivated in MTSV and MTSVY medium for 18 h. Total RNAs were extracted using TRI Regent (SIGMA-Aldrich, USA). Quantitative reverse transcription PCR (RT-qPCR) was carried out to determine the transcriptional levels of the ATPase and TadC proteins. The concentrations of the total RNAs were estimated using the Eppendorf BioSpectrometer® basic (Eppendorf AG, Germany).

First-strand cDNAs were synthesized from the total RNAs according to the protocol from the Maxima First Strand cDNA Synthesis Kit for RT-qPCR with dsDNase (Thermo Scientific, USA). The resulting cDNA preparations were used to evaluate the mRNA levels of the ATPase and TadC proteins, using Luna Universal qPCR Master Mix (New England Biolabs, USA) and gene specific primers (Supp Table 4). qPCR was performed in an Eppendorf MasterCycler RealPlex4 (Eppendorf AG, Germany) with the following steps: denaturing at 95 °C for 2 min, 40 cycles of 95 °C 15 s, 55 °C 15 s and 68 °C 20 s. Relative amounts of mRNAs were evaluated using the comparative Ct method with 16 S rRNA as the reference.

**Cryo-EM conditions and image processing**

The cell appendage extract of *S. islandicus* REY15A grown under poor media (4.5 μl) was applied to glow-discharged lacey carbon grids and then plunge-frozen using an EM GP Plunge Freezer (Leica). The cryo-EMs were collected using EPU on a 300 keV Titan Krios with a K3 camera at 1.08 Å per pixel and a total dose of 50 e⁻/Å². The cryo-EM workflow began with patch motion corrections and CTF estimations in cryoSPARC[48–50]. Next, particle segments were auto-picked by "Filament Tracer" without employing a reference. This step requires the specification of a diameter range, and multiple featureless cylinders of different diameters were generated to serve as references for particle picking. As such, the steps involved were not biased by any pre-existing references or power spectra analysis. All auto-picked particles were subsequently 2D classified with multiple rounds, and all particles in bad 2D averages were removed. Filaments of different forms were assigned into different subsets based on their 2D averages and corresponding power spectrum. After this, the tri-pili dataset had 240,930 particles left with a shift of 16 pixels between adjacent boxes, while the mono-pili dataset had 78,068 particles left with a 6-pixel shift. Next, the possible helical symmetries were calculated from averaged power spectra for each filament form generated from the raw particles. All possible helical symmetries were calculated and then were tested by trial-and-error in cryoSPARC until recognized protein features, such as clear separation of β-strands, the pitch of α-helix and good side chains densities, were observed[51,52]. The volume hand was determined by the

hand of α-helices in the map. After that, 3D reconstruction was performed using "Helical Refinement" first, Local CTF refinement, and then another round of "Helical Refinement" with "Non-uniform Refinement". The resolution of each reconstruction was estimated by Map:Map FSC, Model:Map FSC, and d₉₉[19]. The final volumes were then sharpened with local resolution filtering automatically estimated in cryoSPARC, and the statistics are listed in Table 1.

The cell appendage extract of *S. islandicus* REY15A grown under rich media was frozen under the same condition. 3069 cryo-EM micrographs were collected on a 200 keV Glacios with a Falcon IV camera at 1.5 Å per pixel. The data processing workflow was very similar to the dataset collected on Krios. After 2D classification and analysis of power spectra to validate the absence of tri-pili, this dataset only containing mono-pili was reconstructed to 5.9 Å resolution, as estimated by Map:Map FSC.

**Model building of tri-pili and mono-pili**

The first step of model building is identifying the pilin protein from the experimental cryo-EM map. The pilin protein, produced by two identical genes (SiRe_2654 and SiRe_2659 of *S. islandicus* REY15A), was identified to be the best fit to the EM map among all proteins in this strain by the DeepTracer-ID[20], and direct cryo-EM-based approaches such as ModelAngelo[22] and DeepTracer[23]. Close homologs of the identified pilin were previously reported in the pili structures of *S. islandicus* LAL14/1[6] and *S. solfataricus* POZ149[5]. AlphaFold prediction[21] of the identified pilin was used as the starting model for the model building process. Pilin subunits within an ASU were first docked into the map using ChimeraX[53]. The N-terminal and C-terminal domains were docked separately to accommodate the flexible linker region. Pilin subunits were then manually refined in Coot[54]. Then a filament model was generated and real-space refined in PHENIX[55]. MolProbity was used to evaluate the quality of the filament model[56]. The refinement statistics of both mono-pili and tri-pili are shown in Table 1. Data visualization was primarily done in ChimeraX[53]. Maps used for visualization were sharpened based on local resolution estimation in cryoSPARC. Protein-protein interface, buried surface area and solvent-accessible surface were calculated using PDBePISA[57].

**Semi-quantitative comparisons of pilus abundance**

Given the challenge of distinguishing between mono-pilus and tri-pilus from a single filament on a cryo-EM micrograph, the analysis was conducted in a semi-quantitative manner. The same amount of cells for different strains was harvested before the cryo-EM analysis, and the same shearing and plunge freezing procedure was employed. During the cryo-EM processing, all filaments were auto-picked using the "filament tracer" in cryoSPARC with the same shift distance of 5 Å, and 2D classifications were performed to count particles from distinct filament species. Subsequently, the particle counts were normalized against the WT strain grown under the poor media condition, considering four factors: the camera pixel size; camera width (K3 vs. Falcon 4); total micrographs captured; and sample dilution applied. The comprehensive number of micrographs (N) gathered and the aggregate counted particles prior to normalization (P) for each dataset are illustrated in Fig. 6.

**Reporting summary**

Further information on research design is available in the Nature Portfolio Reporting Summary linked to this article.

## Data availability

The three-dimensional reconstructions have been deposited in the Electron Microscopy Data Bank with accession codes EMD-41286 (mono-pilus) and EMD-41283 (tri-pilus). The atomic models have been deposited in the Protein Data Bank with accession code 8TIF (mono-pilus) and 8TIB (tri-pilus). Source data are provided in this paper.

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

## Acknowledgements

The cryo-EM imaging was done at the Molecular Electron Microscopy Core Facility at the University of Virginia, which is supported by the School of Medicine. This work was supported by NIH grants GM122510 (E.H.E.) and GM138756 (F.W.). The work in the M.K. laboratory was supported by grants from l'Agence Nationale de la Recherche (ANR- 21-CE11-0001-01) and Ville de Paris (Emergence(s) project MEMREMA).

## Author contributions

J.L. and V.C.K. performed sample preparation. G.E., S.T.R., and F.W. performed microscopy image analysis. J.L. generated the knockout strains. F.W., E.H.E., and M.K. obtained funding and supervised the research. G.E., J.L., M.A.B.K. E.H.E., F.W., and M.K. wrote the manuscript with input from all authors.

## Competing interests

The authors declare no competing interests.
