## [Peer Review File · Nature Communications]

Two distinct archaeal type IV pili structures formed by proteins with identical sequenceReviewer #1 (Remarks to the Author):

Two dramatically distinct archaeal type IV pili structures formed by the same pilin by Liu et al

This is an interesting manuscript about pilin structures in *S. islandicus*. While I am in general supportive of this manuscript, there are several points in the manuscript where the authors have not sufficiently backed up their results with experimental data, highlighted below. The major point is that I am not convinced that the protein making the two types of pili is exactly the same either in amino acid sequence or glycosylation. This is the central tenet of the publication, therefore it must be experimentally validated beyond the cryo-EM maps. Right now, this is the biggest drawback of the manuscript.

1. The PDB validation report is the wrong one. It is labeled "Not for peer review". Could the authors provide the correct one?
2. I am not convinced by the "glycans" presented in Fig. 3. Could they not be other chemical species? Could the authors provide mass spectrometry or a reconstruction from a non-glycosylated mutant to prove this?
3. Related to the above, it would be interesting to know the identity of the sugars (if they are indeed sugars). Some biochemical characterisation would be important.
4. The structures presented are at modest resolutions. Therefore, all side chains are not unambiguously identified by the density. I am worried that the mono and tri pilins could be made from mutated proteins. Even a single amino acid mutation could change the outcome of the assembly. Can the authors show peptide fingerprinting mass spec or intact mass spec to support their claims that the protein is the same?
5. Even a small difference in glycosylation could be the reason for different assemblies. I think the authors have to show biochemically that the protein, including glycosylation is the same. In Fig. 3, the glycan densities all look different to my eye. Are these due to differences in noise of the maps? Or actual biochemical differences? This question cannot be answered by cryo-EM maps at this resolution, so this must be further supported by orthogonal experiments. It is absolutely crucial that this is addressed.

Minor points

6. I find a bit strange is that the authors grew the cells in rich media (to show that there are only mono-pile) on a different microscope at different pixel size etc leading to a lower resolution structure (5.9Å) (Methods section). They show the lower resolution 2D classes in the SI along with the others and seem to base their classification on features of the power spectrum. Could the authors elaborate?
7. Regarding the knockouts: Looking at Supp Fig 4, del0180 and del0181 seem to indeed halt pili production. For the other knockouts, their argument seems to be that there are still both mono and triple-pili present, but the ratio seems to change quite a bit (especially in the del1879). Maybe they could comment on that and whether this is relevant (could be due to the significantly different numbers of particles (see caption of Fig 6)).
8. I personally had a hard time understanding Fig 6, could you simplify please?

Reviewer #2 (Remarks to the Author):

In this well-written manuscript from Liu et al, the authors describe two distinctly different cryoEM structures of type 4 pili from *Saccharolobus islandicus* REY15A. Interestingly, these different types

of pili were observed to be expressed under different growth conditions. By reaching a resolution of sub-4Å in their cryoEM reconstructions, it was possible to identify the pilus protein by “reading” the amino acid sequence using two independent approaches. It turned out, that both structures were assembled from the same major pilin protein. The main difference between both structures is the conformation adapted by the major pilin. While in the so-called mono-pilus all subunits are equivalent and the structure resembled previously solved type 4 pilus structures, the pilin protein was found adopting three different conformations in the so-called tri-pilus. These different conformations are facilitated by a flexible linker region between the N-terminal alpha-helix and the C-terminal globular beta-strand-rich domain. The authors further showed that both types of pili are secreted by the same secretion apparatus. However, the expression levels of the involved ATPase and secretion pore were significantly different under the investigated growing conditions.

Liu et al thoroughly describe the intriguing differences between two very diverse pilus structures, which were surprisingly assembled by the same protein. On top, they show effort in corroborating their structural findings through genetic mutations and gene expression analysis. If the authors could deepen their functional insights on the role of the tri-pilus, this manuscript is very well suited for publication in this journal.

Major comments:

- This manuscript was submitted back-to-back on bioRxiv with the manuscript of Gaines et al.. While both groups present similar structures, the main difference is the orientation of the globular C-terminal domain with respect to the N-terminal helix. The authors should show the density of the linker region together with a comprehensive analysis to exclude an error in their model. Without the cryoEM map available, this analysis is not possible for the reviewer.

- Line 156-157: As a genetic system seems to be already available for *S. islandicus*, a deletion of either SiRe_2654 or SiRe_2659 would shine light on the ambiguity of the pili being a product of both genes or only one.

- Is there any difference in the curvature between both types of pili? Are they rather straight or heavily bent? It would be helpful to include 2D cryoEM micrographs for both types of pili in the manuscript.

- This manuscript clearly focuses on the structural aspects of these two type 4 pili and falls a bit short of investigating their function. While extensive functional analysis might be beyond the scope of this manuscript, some more insights would help to narrow down the speculation on the function. For example, can the movement of these cells be analyzed in full and low nutrient medium? Is the tri-pilus involved in twitching motility similar to the Aap in *Sulfolobus acidocaldarius*?

Minor comments:

- Title: The word “dramatically” in the title should be toned down.

- Line 100-102: The authors directly started with their cryoEM results. It would help the reader if one sentence could be added about their sample preparation method.

- Line 142: What means “semiquantitative comparisons of pilus abundance”? How was this calculated? I unfortunately could not find any details in the method section.

- Glycosylation analysis: While the authors describe O-glycosylation sites, it is unclear to me if there are also any N-glycans.

- The bioRxiv paper from Gaines et al. should be cited.

Reviewer #3 (Remarks to the Author):

This manuscript describes two forms of type IV pili (T4P) from the archeon *S.islandicus*. The novel discovery is that their helical assemblies differ depending on the richness of the growth media, while the protein folds of their subunits remain essentially identical. While the mono-pilus with classical T4P architecture is prevalent at rich media conditions, the tri-pilus results from lean media and features 3 distinct 180 degrees-rotated orientations of the outer domain in the helical asymmetric unit. Both assemblies retain nearly identical core domains.

The work is significant because it is the first instance that such domain rotation and multiple conformations within the asymmetric unit has been reported in archaeal pili (a recent paper by Daum et al also found similar multi-conformational arrangement in a different pilus). What makes this paper stand out is a proper helical analysis that leaves no doubt about the two symmetries. Using extensive mutagenesis, the authors further demonstrated that both pili are products of the same secretion system and provide a compelling hypothesis about the biological significance of this differential expression system governed by the growth conditions. The latter is part of the discussion and is clearly presented as best attempt to interpret the observations; proof of which is beyond the scope of their already comprehensive insights.

The experiments are complete and support the conclusions by the authors.

The important helical analysis is done thoroughly, as expected from these experienced collaborators.

Map and model validation reports ascertain good quality (with a few sidechain outliers in the outer domains which are insignificant)

The manuscript is written well and presented clearly. The figures are excellent.

Minor points of critique:

1. The use of the terms pilus/pili and filament is not always consistent, e.g. the abstract talks about both forms of pili requiring the same ATPase and TadC-like pore, indicating that the same secretion system can produce different filaments (lines 44-45) – in this case, use “different pili”, since this paper makes explicit claims about filaments vs pili from a structural point of view. Please check all occurrences in the manuscript.
2. If the semi-quantitative comparison (Line 142) for cells grown in MTSV medium corresponds to “WT (poor)” in SI Fig.4, the stated value of “tri-pili 6.5x more abundant” should be consistent with the 2D class distribution. The red labels suggest 1 out of 9 (87%) were mono-pili among tri-pili. Please adjust the value or refer to the relevant strains or to the data in Fig.6b. Fig.6 legend (line 550) “semi-quantification” > “semi-quantitative”
3. There is no description in the methods or legend how the extra density attributed to glycosylation was visualized (Fig.3). Is this a difference map between model-generated and experimental maps, or manual assignment and coloring of the isosurfaces? Please add a brief paragraph about the procedure to the methods. How is the “display threshold [%]” defined? The contour level is usually given as standard deviation above average (after normalizing the map to average 0 and standard deviation 1). Which Ser and Thr exhibit extra density? Is there any evident N-linked glycosylation?
4. Contour levels of the normalized maps should be indicated in the legends of all figures that show iso-contoured surfaces of cryoEM maps. Scale bars should be added to most figures. Where appropriate, please indicate in the figure legends if the visualized maps were sharpened.
5. Line 457: “5.9 A resolution” is likely a typo and should be 3.9 A.
6. Line 288: “we always see 3 different strands...” – of course, because these particles were picked according to their power spectra, and symmetrized. Random patches could have ended up in junk

classes. Suggest to remove this sentence.

7. How were the solvent-accessible surfaces calculated and visualized (Fig.4 and SI Fig.3)? Please add description to the methods.

Reviewers' Comments:

Reviewer #1 (Remarks to the Author)

Two dramatically distinct archaeal type IV pili structures formed by the same pilin by Liu et al

This is an interesting manuscript about pilin structures in *S. islandicus*. While I am in general supportive of this manuscript, there are several points in the manuscript where the authors have not sufficiently backed up their results with experimental data, highlighted below. The major point is that I am not convinced that the protein making the two types of pili is exactly the same either in amino acid sequence or glycosylation. This is the central tenet of the publication, therefore it must be experimentally validated beyond the cryo-EM maps. Right now, this is the biggest drawback of the manuscript.

We agree with the reviewer that we cannot definitively establish that the glycosylation levels are identical between mono- and tri-pili. Accordingly, we have toned down our arguments in that section and allowed for the possibility that this might be the case. However, as for the amino acid sequence of the pilin, we confirmed that they are identical through re-sequencing both pilin gene copies present in the genome. We also added another supplemental figure to show that no other sequences in the genome could be fit into the EM density.

(1) 1. The PDB validation report is the wrong one. It is labeled "Not for peer review". Could the authors provide the correct one?

We have updated the validation reports.

2. I am not convinced by the "glycans" presented in Fig. 3. Could they not be other chemical species? Could the authors provided mass spectrometry or a reconstruction from a non-glycosylated mutant to prove this?

Similar glycan patterns have been noted in two previously reported mono-pili studies (Wang et al., 2020; Wang et al., 2019). Mass spectrometry is not particularly useful in this scenario given the uncertainty regarding the specific glycan species involved. In a closely related strain, *Sulfolobus islandicus* LAL14/1, despite utilizing a triple enzyme digestion approach, we could only identify a single peptide fragment from *Sulfolobus islandicus* LAL14/1 mono pili. Therefore, we employed scanning transmission electron microscopy (STEM) to estimate that post-translational modification, presumably glycosylation, contributes approximately 35% of the molecular weight of the *Sulfolobus islandicus* LAL14/1 mono-pili (Wang et al., 2019). This ~35% extra mass can be further removed by a trifluoromethane sulfonic acid (TFMS) deglycosylation kit. Finally, the mechanism and enzymes involved in O-glycosylation in Sulfolobales are not known, precluding contemplation of non-glycosylated mutants.

3. Related to the above, it would be interesting to know the identity of the sugars (if they are indeed sugars). Some biochemical characterisation would be important.

We share the reviewer's interest in identifying the specific sugars involved, but such an investigation is beyond the scope of the current manuscript. The study of O-linked glycosylation

presents a myriad of challenges, encompassing glycan heterogeneity—both in terms of the monosaccharide units and their respective linkages and branching—as well as the difficulty in discerning glycan isomers, intricate sample preparation for glycan enrichment and purification, and a general dearth of specialized tools for these studies, among other issues. Additionally, there is no universally recognized sequon for O-glycosylation, save for the occurrence at serine and threonine residues. In our study, all the extra densities observed were located at these specific residues. While this does not conclusively establish that these densities are 100% sugars, there are, to our knowledge, no other post-translational modifications (PTMs) that exclusively occur at these two residues. In prior research on the *Sulfolobus islandicus* LAL14/1 mono-pilus, which shares 96% sequence identity with the pilus described in this manuscript, we demonstrated that the additional molecular weight attributed to post-translational modification could be eliminated using a trifluoromethane sulfonic acid (TFMS) deglycosylation kit, which also strongly suggests they are O-linked sugars.

4. The structures presented are at modest resolutions. Therefore, all side chains are not unambiguously identified by the density. I am worried that the mono and tri pilins could be made from mutated proteins. Even a single amino acid mutation could change the outcome of the assembly. Can the authors show peptide fingerprinting mass spec or intact mass spec to support their claims that the protein is the same?

We confirmed two copies of the pilin gene encode identical proteins through Sanger sequencing of the corresponding genes. This is now mentioned in the revised text. Furthermore, we have incorporated an additional supplemental figure to demonstrate that no other sequences in the genome agree with the observed EM density. The use of EM for protein identification is now a routine practice, and the resolution achieved in this study (~ 3.5 Å) is more than adequate for this purpose (Chang et al., 2022; Jamali et al., 2023). As elaborated above, mass spectrometry will not have enough sequence coverage in this specific context due to the extensive O-glycosylation present, which cannot be easily removed with commercially available enzymes. Further, we have found no conditions where only the tri-pilus is present and the mono-pilus is absent. Thus, there is no bulk assay (such as mass spectrometry) that can give results distinguishing the two.

5. Even a small difference in glycosylation could be the reason for different assemblies. I think the authors have to show biochemically that the protein, including glycosylation is the same. In Fig. 3, the glycan densities all look different to my eye. Are these due to differences in noise of the maps? Or actual biochemical differences? This question cannot be answered by cryo-EM maps at this resolution, so this must be further supported by orthogonal experiments. It is absolutely crucial that this is addressed.

As already mentioned above, we agree with the reviewer that we cannot definitively establish that the glycosylation levels are identical between mono- and tri-pili. What we can establish is that the amount of additional post-translational modification (PTM) density appears similar. The variations in glycan densities may seem different to the eye arising from filtering at different resolutions (~ 3.5 Å vs. ~ 4.0 Å). In light of this, we have updated the glycosylation figure to display glycosylation at the same resolution for both, filtered to 7 Å. We also added the projections of pili reconstructions at high and low thresholds for mono and tri-pili, to show

the extent of glycosylation is similar. Nevertheless, we have toned down our arguments in that particular section and allow for the possibility that the differences could be explained by differences in glycosylation.

Minor points

6. I find a bit strange is that the authors grew the cells in rich media (to show that there are only mono-pili) on a different microscope at different pixel size etc leading to a lower resolution structure (5.9Å) (Methods section). They show the lower resolution 2D classes in the SI along with the others and seem to base their classification on features of the power spectrum. Could the authors elaborate?

The lower 5.9 Å resolution was likely due to a limited number of particles and specific microscope settings. It is generally accepted that the Krios microscope offers advantages over the Glacios, and that the presence of an energy filter is preferable. In this 5.9 Å dataset, neither a Krios microscope nor an energy filter was used, largely contributing to the lower resolution obtained. Given that we had already obtained a ~4 Å resolution map for the mono-pilus and could adequately differentiate between tri- and mono-pili at lower resolutions through power spectra and 2D averages, we did not feel it necessary to invest an additional \$4,000 for data collection on the mono-pilus structure that we already have. As a result, we opted for a more cost-effective approach, gathering a smaller dataset using the Glacios microscope at a cost of ~\$500.

7. Regarding the knockouts: Looking at Supp Fig 4, del0180 and del0181 seem to indeed halt pili production. For the other knockouts, their argument seems to be that there are still both mono and triple-pili present, but the ratio seems to change quite a bit (especially in the del1879). Maybe they could comment on that and whether this is relevant (could be due to the significantly different numbers of particles (see caption of Fig 6)).

While we did note a difference in the ratio, we are cautious not to overinterpret this finding. Cryo-EM is not ideally suited for quantitative analysis of ratios of two different forms, and variability can occur when imaging different areas of the grid. For example, one form of the pilus might preferentially adsorb to the carbon film or the air-water interface. Therefore, we find it more appropriate to employ cryo-EM for making binary (yes-or-no) observations or for semi-quantitative assessments as presented in this paper.

8. I personally had a hard time understanding Fig 6, could you simplify please?.

We have simplified the figure by adding labels and explaining the knockouts and hopefully now this figure is less confusing.

Reviewer #2

In this well-written manuscript from Liu et al, the authors describe two distinctly different cryoEM structures of type 4 pili from *Saccharobolus islandicus* REY15A. Interestingly, these

different types of pili were observed to be expressed under different growth conditions. By reaching a resolution of sub-4Å in their cryoEM reconstructions, it was possible to identify the pilus protein by “reading” the amino acid sequence using two independent approaches. It turned out, that both structures were assembled from the same major pilin protein. The main difference between both structures is the conformation adapted by the major pilin. While in the so-called mono-pilus all subunits are equivalent and the structure resembled previously solved type 4 pilus structures, the pilin protein was found adopting three different conformations in the so-called tri-pilus. These different conformations are facilitated by a flexible linker region between the N-terminal alpha-helix and the C-terminal globular beta-strand-rich domain. The authors further showed that both types of pili are secreted by the same secretion apparatus. However, the expression levels of the involved ATPase and secretion pore were significantly different under the investigated growing conditions.

Liu et al thoroughly describe the intriguing differences between two very diverse pilus structures, which were surprisingly assembled by the same protein. On top, they show effort in corroborating their structural findings through genetic mutations and gene expression analysis. If the authors could deepen their functional insights on the role of the tri-pilus, this manuscript is very well suited for publication in this journal.

We thank the reviewer for the comments.

Major comments:

- This manuscript was submitted back-to-back on bioRxiv with the manuscript of Gaines et al.. While both groups present similar structures, the main difference is the orientation of the globular C-terminal domain with respect to the N-terminal helix. The authors should show the density of the linker region together with a comprehensive analysis to exclude an error in their model. Without the cryoEM map available, this analysis is not possible for the reviewer.

We have now added a figure to the Supplement showing the clearly resolved linker. The reviewer is raising the possibility that the C-terminal domain of one subunit may be incorrectly attached in our model to the N-terminal domain of a different subunit, but the linker density shows that this is not possible. But even if such an error had been made, it would generate a C-terminal domain that was substantially translated with respect to the N-terminal domain – it could not generate a C-terminal domain that was rotated nearly 180° in orientation, and be almost upside-down with respect to the domain in the mono-pilus.

- Line 156-157: As a genetic system seems to be already available for *S. islandicus*, a deletion of either SiRe_2654 or SiRe_2659 would shine light on the ambiguity of the pili being a product of both genes or only one.

This is a good suggestion and we have already considered the feasibility of such experiment. Unfortunately, because the two pilin genes are identical, including the cognate promoter regions, the efficient CRISPR-based gene knockout technology, used to delete all ATPase and TadC-like genes described in the manuscript, cannot be used.

- Is there any difference in the curvature between both types of pili? Are they rather straight or heavily bent? It would be helpful to include 2D cryoEM micrographs for both types of pili in the manuscript.

We had already incorporated 2D averages for all the collected datasets in Supplemental Figure, and we do not observe notable curvature in these pili which would appear in the 2D averages if it were present. We have now added some representative micrographs to the Supplement showing that the flexibility of the two pili appears to be similar.

- This manuscript clearly focuses on the structural aspects of these two type 4 pili and falls a bit short of investigating their function. While extensive functional analysis might be beyond the scope of this manuscript, some more insights would help to narrow down the speculation on the function. For example, can the movement of these cells be analyzed in full and low nutrient medium? Is the tri-pilus involved in twitching motility similar to the Aap in *Sulfolobus acidocaldarius*?

We agree that understanding the function of the mono- and tri-pili is an important future research direction, but such functional studies are beyond the scope of the present manuscript. Unfortunately, we do not have access to a technical setup to perform the twitching motility experiment described for *S. acidocaldarius*. We note that the latter is a standalone study, separate from the structure of the *S. acidocaldarius* pili structure. Importantly, one might wonder whether the conformation of the pili produced under the conditions used for twitching motility experiment (i.e., live-cell imaging in small volumes with low aeration) is the same as that determined for the cells grown in a flask under vigorous aeration.

Minor comments:

- Title: The word “dramatically” in the title should be toned down.

We have removed “dramatically” from the title.

- Line 100-102: The authors directly started with their cryoEM results. It would help the reader if one sentence could be added about their sample preparation method.

We have now added such a transition sentence.

- Line 142: What means “semiquantitative comparisons of pilus abundance”? How was this calculated? I unfortunately could not find any details in the method section.

More details have now been added into the Methods section.

- Glycosylation analysis: While the authors describe O-glycosylation sites, it is unclear to me if there are also any N-glycans.

There are no extra densities observed associated with the four asparagine side chains present in this pilin. We have added a sentence explaining this.

- The bioRxiv paper from Gaines et al. should be cited.

Done

Reviewer #3

This manuscript describes two forms of type IV pili (T4P) from the archeon *S.islandicus*. The novel discovery is that their helical assemblies differ depending on the richness of the growth media, while the protein folds of their subunits remain essentially identical. While the monopilus with classical T4P architecture is prevalent at rich media conditions, the tri-pilus results from lean media and features 3 distinct 180 degrees-rotated orientations of the outer domain in the helical asymmetric unit. Both assemblies retain nearly identical core domains.

The work is significant because it is the first instance that such domain rotation and multiple conformations within the asymmetric unit has been reported in archaeal pili (a recent paper by Daum et al also found similar multi-conformational arrangement in a different pilus). What makes this paper stand out is a proper helical analysis that leaves no doubt about the two symmetries. Using extensive mutagenesis, the authors further demonstrated that both pili are products of the same secretion system and provide a compelling hypothesis about the biological significance of this differential expression system governed by the growth conditions. The latter is part of the discussion and is clearly presented as best attempt to interpret the observations; proof of which is beyond the scope of their already comprehensive insights.

The experiments are complete and support the conclusions by the authors.

The important helical analysis is done thoroughly, as expected from these experienced collaborators.

Map and model validation reports ascertain good quality (with a few sidechain outliers in the outer domains which are insignificant)

The manuscript is written well and presented clearly. The figures are excellent.

We appreciate the kind words from the reviewer.

Minor points of critique:

1. The use of the terms pilus/pili and filament is not always consistent, e.g. the abstract talks about both forms of pili requiring the same ATPase and TadC-like pore, indicating that the same secretion system can produce different filaments (lines 44-45) – in this case, use “different pili”, since this paper makes explicit claims about filaments vs pili from a structural point of view. Please check all occurrences in the manuscript.

We have now revised the paper to make the terms consistent.

2. If the semi-quantitative comparison (Line 142) for cells grown in MTSV medium corresponds to “WT (poor)” in SI Fig.4, the stated value of “tri-pili 6.5x more abundant” should be consistent with the 2D class distribution. The red labels suggest 1 out of 9 (87%) were mono-pili among tri-pili. Please adjust the value or refer to the relevant strains or to the data in Fig.6b.

Fig.6 legend (line 550) “semi-quantification” > “semi-quantitative”

We have now revised the text as suggested, and added explanation that only first 10 most abundant 2D classes were shown. In total there are 50 classes, and after the first 10, there are more classes containing carbon edges etc. Also each class may have different number of

particles, and that 6.5x was calculated after combining all particles of tri-pili and mono-pili classes first, so it is accurate and doesn't need to be changed.

3. There is no description in the methods or legend how the extra density attributed to glycosylation was visualized (Fig.3). Is this a difference map between model-generated and experimental maps, or manual assignment and coloring of the isosurfaces? Please add a brief paragraph about the procedure to the methods. How is the "display threshold [%]" defined? The contour level is usually given as standard deviation above average (after normalizing the map to average 0 and standard deviation 1). Which Ser and Thr exhibit extra density? Is there any evident N-linked glycosylation?

We have now explained in the figure legend how the extra density was visualized. No N-linked glycosylation was observed and we have added this into the Results. In cryo-EM, density maps are NOT usually normalized to a mean (e.g., 0.0) and a standard deviation (e.g., 1.0). While this is usual in x-ray crystallography, maps have a different basis than those generated by cryo-EM. For example, F_{000} is not included in the Fourier synthesis, thus a map will have zero mean. But there is absolutely no reason why a Coulomb potential map should have zero mean. Further, arguments have been made in crystallography why the choice of a scale based on σ (the standard deviation) is a poor choice (Lang et al., 2014): "it is difficult to determine which density features are signal vs. noise because the σ unit has little to do with the uncertainty in the electron density. It is also impossible to quantitatively compare features in different maps, because the scale and offset relating σ to the absolute electron density varies among crystals of different molecules or even of the same molecular species with different symmetries or crystallization solvents." There are actually many more reasons for this normalization not being done in cryo-EM, having to do with different microscope and camera transfer functions, the use of different software packages that produce maps in different ways, alignment errors, etc. The main tool used by most people in cryo-EM for displaying maps is Chimera (or ChimeraX), and an online tutorial for Chimera states: "Electron microscopy density maps are usually not normalized (e.g. mean = 370.51, SD = 1477.11), scale is not meaningful."

4. Contour levels of the normalized maps should be indicated in the legends of all figures that show iso-contoured surfaces of cryoEM maps. Scale bars should be added to most figures. Where appropriate, please indicate in the figure legends if the visualized maps were sharpened.

We have now added most of this information as suggested into figures. As explained above, contour levels of cryo-EM maps are on arbitrary scales, and such information about the arbitrary threshold used is not normally given in figure legends.

5. Line 457: "5.9 A resolution" is likely a typo and should be 3.9 A.

This 5.9 Å map was collected on a Glacios microscope without an energy filter, using a limited number of segments. As discussed in response to a previous comment, since we had already obtained a ~4 Å resolution map for the mono-pilus and could adequately differentiate between tri- and mono-pili at lower resolutions through power spectrum and 2D averages, we did not find it necessary to invest an additional \$4,000 for data collection on the mono-pilus structure that we already had. As a result, we opted for a more cost-effective approach, gathering a smaller dataset on the Glacios microscope at a cost of ~ \$500.

6. Line 288: “we always see 3 different strands...” – of course, because these particles were picked according to their power spectra, and symmetrized. Random patches could have ended up in junk classes. Suggest to remove this sentence.

In fact, all particles were selected without employing a reference, utilizing the filament tracer function in cryoSPARC. This option in cryoSPARC requires the specification of a diameter range, and the software generates featureless cylinders of different diameters to serve as references for particle picking. As such, the steps involved were not biased by any pre-existing references or power spectra analysis. We have now added these details to the Methods.

7. How were the solvent-accessible surfaces calculated and visualized (Fig.4 and SI Fig.3)? Please add description to the methods.

Those calculations were done in PISA and this has now been added into the Methods.

References

Chang, L., Wang, F., Connolly, K., Meng, H., Su, Z., Cvirkaite-Krupovic, V., Krupovic, M., Egelman, E.H., and Si, D. (2022). DeepTracer-ID: De novo protein identification from cryo-EM maps. *Biophys. J.* *121*, 2840-2848.

Jamali, K., Käll, L., Zhang, R., Brown, A., Kimanius, D., and Scheres, S.H.W. (2023). Automated model building and protein identification in cryo-EM maps. *bioRxiv*, 2023.2005.2016.541002.

Lang, P.T., Holton, J.M., Fraser, J.S., and Alber, T. (2014). Protein structural ensembles are revealed by redefining X-ray electron density noise. *Proc. Natl. Acad. Sci. U.S.A.* *111*, 237-242.

Wang, F., Baquero, D.P., Su, Z., Beltran, L.C., Prangishvili, D., Krupovic, M., and Egelman, E.H. (2020). The structures of two archaeal type IV pili illuminate evolutionary relationships. *Nat. Commun.* *11*, 3424.

Wang, F., Cvirkaite-Krupovic, V., Kreutzberger, M.A.B., Su, Z., de Oliveira, G.A.P., Osinski, T., Sherman, N., DiMaio, F., Wall, J.S., Prangishvili, D., Krupovic, M., and Egelman, E.H. (2019). An extensively glycosylated archaeal pilus survives extreme conditions. *Nat Microbiol* *4*, 1401-1410.

Reviewer #1 (Remarks to the Author):

Review for Liu et al

I have real respect for these authors' previous work, therefore I am disappointed that they have decided not to provide any experimental validation of the structural differences beyond the original cryoEM. Some points to mention -

1. Just because mass spectrometry was difficult in another pilin molecule, it is not a guarantee that it will be the same scenario in this situation. Why not do the experiment and answer with the observations?

2. I am afraid $\sim 3.5 \text{ \AA}$ is not sufficient resolution to ID each side chain, this is weak argument.

3. Why filter both structures to 7 \AA ? This seems very arbitrary. If you are comparing a 3.5 \AA with a 4 \AA resolution structure, the correct resolution to filter both structures is 4 \AA (and not 7 \AA). My suspicion is that the authors tested many filters, and at 7 \AA , they could get the densities to look similar, and this is why they chose this. Unfortunately I am not convinced.

In summary, if the authors can provide convincing experimental backing of their claims, I would be delighted. But if they are not interested in doing any further experiments, then I strongly recommend removing the words "from the same pilin" from the manuscript title. This is absolutely not proven in my opinion.

Reviewer #2 (Remarks to the Author):

The authors have made an effort to address the concerns and the revised version of the manuscript has been improved according to the reviewers' recommendations.

However, I still think that investigating the ambiguity of the two pili being either a product of one or both of the genes SiRe_2654 and SiRe_2659 would be desirable. I understand, that not being able to use the efficient CRISPR-based gene knockout technology is a pity, but are there no other more conventional cloning approaches for *S. islandicus*?

Reviewer #3 (Remarks to the Author):

The authors have sufficiently addressed all relevant points of critique.

I look forward to seeing this interesting paper in print.

Responses to comments:

We appreciate new comments from the reviewers. However, we think the mass spectrometry analysis and gene knockouts requested by Reviewer #1 and Reviewer #2, respectively, are problematic and unlikely to provide the definitive answers expected by the reviewers. As we explained in the previous rebuttal letter, under no conditions could we obtain a preparation which would contain exclusively tri-pili. Instead, tri-pili were present in a mixture with mono-pili. Therefore, given that the pilin sequences are the same between mono- and tri-pili, a bulk assay such as mass spectrometry would not distinguish which peptide fragment is coming from which filament form. Also, the pilin has very limited digestion sites and residues are shielded by glycosylation from such digestions. A tri-enzymatic (chymotrypsin, trypsin and Asp N) digestion of LAL14/1 pili (96% sequence identity to the pilin in this paper) yielded a single peptide fragment detectable by mass spec (1). Analysis of a single peptide would hardly provide the desired information which would allow the distinction between the mono- and tri-pili. Further, the genes have been re-sequenced, showing that they encode identical proteins. If the point of the mass spectrometry is to look at glycosylation, even if one can detect the glycosylation species (which is unlikely to be the case here), how does one relate the glycosylation pattern to a particular filament form. Since we now state that the difference between the two filament forms could conceivably involve differences in glycosylation, what would we learn if we see variability in glycosylation (e.g., the same peptide that is glycosylated and not glycosylated)? This would not show that the potential difference in glycosylation causes the conformational change. We disagree with the reviewer that we do not provide sufficient evidence that the two forms of filaments are produced from the same pilin. We have tested AlphaFold models of all proteins from the *S. islandicus* proteome and the pilin encoded by SiRe_2654/SiRe_2659 was the only one which fit the electron density map perfectly. Thus, the pilins in both filaments have the same sequence and hence are the same protein.

With regard to the gene knockouts, we agree with the reviewer that it would be interesting to knock out either of the two identical pilin genes. However, suppose we knock out SiRe_2654 or SiRe_2659 and observe that only one form now exists (the idealized situation), the mono-pilus or the tri-pilus. What insight does that give us into why there are two pilus forms? Both genes are encoding the same protein, as confirmed by resequencing. So we could spend months (it takes 6-9 months to generate mutants in *S. islandicus* using the pop-in/pop-out strategy) doing this work and observe no difference with the knockouts, or observe a difference that provides no explanation for the multiple states. Thus, we believe that the expenditure of the resources and time for this pursuit is unwarranted.

Lastly, we have observed four conformations for the pilin (three in the tri-pilus and one in the mono-pilus). Reviewer #1 appears willing to accept that three conformations are possible from the same protein, but the fourth one may be due to a different protein. We cannot understand the logic here.

Reviewer #1:

1. Just because mass spectrometry was difficult in another pilin molecule, it is not a guarantee that it will be the same scenario in this situation. Why not do the experiment and answer with the observations?

As stated above, there is 96% sequence identity between the “difficult” pilin and the present one. Would one realistically expect that they would behave completely differently? For the proposed mass spectrometry, looking at the glycosylation would be pointless given the likely heterogeneity of glycans at the same position in different pilin subunits and inability to separate mono-pili and tri-pili, as discussed above.

2. I am afraid ~3.5 Å is not sufficient resolution to ID each side chain, this is weak argument.

Correct, 3.5 Å is not sufficient to ID each side chain, and we have never made an argument otherwise. But what we stated in our previous Response is still true: “We also added another supplemental figure to show that no other sequences in the genome could be fit into the EM density.” So one does not need to ID each side chain, because the possible sequences are finite, denumerable, and encoded by the genome. It is not as if we need to determine whether a specific residue is alanine or glycine, both small side chains. If the only protein sequence that fits in the map has a glycine at that position, then one can say with high confidence that this residue must be a glycine.

3. Why filter both structures to 7 Å? This seems very arbitrary. If you are comparing a 3.5 Å with a 4 Å resolution structure, the correct resolution to filter both structures is 4 Å (and not 7 Å). My suspicion is that the authors tested many filters, and at 7 Å, they could get the densities to look similar, and this is why they chose this. Unfortunately I am not convinced.

We have added a supplemental figure with filtrations to 4, 5 and 6 Å. No conclusions are changed.

In summary, if the authors can provide convincing experimental backing of their claims, I would be delighted. But if they are not interested in doing any further experiments, then I strongly recommend removing the words “from the same pilin” from the manuscript title. This is absolutely not proven in my opinion.

We have changed the title to “Two distinct archaeal type IV pili structures formed by proteins of the same sequence” which should satisfy the reviewer.

Reviewer #2

The authors have made an effort to address the concerns and the revised version of the manuscript has been improved according to the reviewers' recommendations.

However, I still think that investigating the ambiguity of the two pili being either a product of one or both of the genes SiRe_2654 and SiRe_2659 would be desirable. I understand, that not being able to use the efficient CRISPR-based gene knockout technology is a pity, but are there no other more conventional cloning approaches for *S. islandicus*?

We agree that this would be an interesting question, but one must weigh the benefit or insights that might be gained from such an approach that would take many months. As discussed above, knowing that when one of these two genes is knocked out only one filamentous form exists would still tell us nothing about why this is the case given that both encode the same protein sequence.

1. F. Wang *et al.*, An extensively glycosylated archaeal pilus survives extreme conditions. *Nat Microbiol* **4**, 1401-1410 (2019).